# Can Large Vision-Language Models Correct Grounding Errors by Themselves?

## Abstract

Enhancing semantic grounding abilities in Vision-Language Models (VLMs) often involves collecting domain-specific training data, refining the network architectures, or modifying the training recipes. In this work, we venture into an orthogonal direction and explore semantic grounding in VLMs through *self-correction*, without requiring in-domain data, fine-tuning, or modifications to the network architectures. Despite the concerns raised in the self-correction of LLMs, we find that if prompted and framed properly, VLMs *can* correct their own semantic grounding mistakes even without the access to the oracle feedback. We also show an identified self-correction framework in an iterative setting which *consistently* improves performance across all models investigated. Overall, iterative self-correction consistently improves VLM performance by up to 8.4 accuracy points across all models investigated; yet, after several rounds of feedback, strong models like GPT-4V and GPT-4o still exhibit significant error rates, indicating promising directions for further research.

## 1 Introduction

The evolution of Large Language Models (LLMs) to encompass multimodal inputs has given rise to an emerging paradigm of general-purpose models that can solve multimodal understanding problems via user-prompt interaction (Touvron et al., 2023; Team et al., 2024; 2023; Yang et al., 2023b; McKinzie et al., 2024). Vision-Language Models (VLMs) are a growing family of multimodal models that simultaneously understand both visual and language cues. These models have demonstrated strong zero-shot performance on tasks including image classification (Deng et al., 2009), captioning (Young et al., 2014), visual question answering (Antol et al., 2015; Goyal et al., 2017), reasoning (Yu et al., 2016; Yuksekgonul et al., 2023) and robotics (Cui et al., 2024; Nasiriany et al., 2024b).

Despite VLMs' strong visual-language understanding abilities, fine-grained visual grounding remains a challenge. Specifically, VLMs struggle to understand region-specific information within complex scenes, for example, when the models are prompted to describe specific objects within a crowded image (Chen et al., 2023; Yang et al., 2023a; You et al., 2023) (See Fig. 1). Prior works address this limitation with additional in-domain data (Guo et al., 2024; Lin et al., 2023; Li et al., 2023), finetuning, or architectural changes (Li et al., 2024; Liu et al., 2024). However, these approaches demand considerable cost in compute (Cai et al., 2023; You et al., 2023). Therefore, enhancing VLMs for fine-grained visual grounding without significant computational overhead remains a challenge.

On the other hand, the adjacent LLMs literature has demonstrated that LLMs can correct their own mistakes (Madaan et al., 2024; Shinn et al., 2023), suggesting a potential way to improve VLMs without additional training. This behavior is coined as *self-correction*, a framework that refines responses from LLMs using LLMs during inference, possibly with external tools or knowledge (Chen et al., 2024; Gou et al., 2024). However, follow-up works in LLMs Kamoi et al. (2024); Huang et al. (2023) argue that LLMs struggle to self-correct without the access to *oracle feedback*. Up to now, there is no clear consensus on when LLMs can effectively perform self-correction (Kamoi et al., 2024). Prior work suggests self-correction is limited by feedback quality (Gou et al., 2024; Olausson et al., 2024) and is more reliable with tools like search engines or compilers (Huang et al., 2023).

In this work, we explore self-correction in VLMs with a focus on multi-modal understanding connecting language to visual concepts—a largely unexplored area to date. Specifically, we investigate self-correction within semantic grounding tasks, as illustrated in Fig. 1 . Semantic grounding is well-

Figure 1: **Enhancing semantic grounding in VLMs through self-correction.** We explore to improve semantic grounding in VLMs through self-correction, without the needs of in-domain data, fine-tuning, or architectural changes. For self-correction, we adopt the setup involving explicit feedback generation. When provided with an image and a specified region, a VLM identifies the semantic properties of the image region. An automated feedback-based verification mechanism facilitates an interaction between the VLM and a 'Verifier' to improve the VLM's initial understanding.

suited for this exploration because it demands the integration of language and visual concepts, requires fine-grained visual understanding, and involves multi-modal reasoning, all of which have significant real-world applications as well as the task itself (Vasudevan et al., 2018; Mitchell et al., 2013; Deruyttere et al., 2019). More importantly, VLMs have demonstrated the ability to provide useful feedback in some visual tasks (Lu et al., 2024; Zhang et al., 2023a), leaving the door open for self-correction in VLMs. Specifically, we focus on two key questions: **(Q1)** Can VLMs receive and understand grounding feedback? **(Q2)** Can VLMs provide grounding feedback? We then combine the key findings from Q1 and Q2 to evaluate whether VLMs can self-correct their mistakes by leveraging another instance of the same model during inference. To mitigate the high difficulty of generating reliable feedback, we identify that semantic grounding can be *decomposed* into easier binary verification tasks, therefore, getting more reliable feedback.

We evaluate the effectiveness of self-correction in our context by repurposing panoptic segmentation datasets from ADE20k (Zhou et al., 2017) and COCO (Lin et al., 2014) for semantic grounding (Yang et al., 2023a; Zhang et al., 2024). We analyze three state-of-the-art open-source VLMs (LLaVA-1.5 Liu et al. (2023a), ViP-LLaVA Cai et al. (2024), and CogVLM Wang et al. (2024)) and two proprietary VLMs (GPT-4V Yang et al. (2023b) and GPT-4o) to identify consistent trends. Finally, *with no additional finetuning and no access to the oracle feedback*, we show that the self-correction framework improves semantic grounding performance in VLMs by up to 8.4 accuracy points.

Below, we summarize the key findings in our exploration:

**1. VLMs can receive and understand feedback to improve semantic grounding.** With a single round of oracle binary feedback, open-source VLMs improve their semantic grounding performances up to 9 accuracy points, suggesting the feedback potentials to improve grounding performance in VLMs (Sec. 4.1).

**2. VLMs can provide high-quality feedback for themselves.** By decomposing semantic grounding into an easier binary verification step and adopting visual prompts, the identified binary verification mechanism improves feedback quality up to an 18-point in $F_1$ score compared to the baseline (Sec. 4.2).

**3. Under the iterative self-correction framework, VLMs improve semantic grounding accuracy up to 8.4 accuracy points *without* the access to the oracle.** Across five VLMs, including three open-source and two proprietary, GPT-4V and GPT-4o, our findings *consistently* indicate that feedback enhances semantic grounding in VLMs (Sec. 5.2).

**4. Open-source VLMs make errors in semantic grounding even if feedback explicitly states the ground truths.** The fact that some models could fail in approximately 25% of cases in this scenario highlights a deficiency in prompt-following capabilities that should be investigated further (Sec. 4.1).

**5. Strong proprietary VLMs show significant improvement but still retain limited capability in leveraging ground-truth oracles**. After three rounds of binary oracle feedback, GPT-4V and GPT-4o improve grounding accuracy substantially but still maintain error rates above 40% on the ADE20k dataset (Sec. 5.2).

## 2 RELATED WORK

**Self-Correction in LLMs:** LLMs have shown some ability to criticize, refine, and correct their responses through prompt-based feedback (Kim et al., 2023; Madaan et al., 2023; Gou et al., 2024), supervised finetuning (Havrilla et al., 2024; Zelikman et al., 2022; Singh et al., 2024) or reinforcement learning (Kumar et al., 2024). This work examines whether VLMs can self-correct via prompt-based feedback. There remains little consensus on whether LLMs can effectively self-correct through additional prompts (Havrilla et al., 2024). While previous studies suggest promise in prompt-based self-correction, they often rely on oracle feedback (Kim et al., 2023; Shinn et al., 2023), or weak prompts for initial responses (Madaan et al., 2023; Bai et al., 2022). Follow-up research suggests that feedback generation limits self-correction (Havrilla et al., 2024). On the other hand, prompt-based self-correction generally excels when useful external tools, such as code executors or search engines, are accessible (Huang et al., 2023; Chen et al., 2024; Gou et al., 2024; Gao et al., 2023); however, these tools are often unavailable in many scenarios. Fact-checking also shows success, as demonstrated by CoVe, which decomposes generation tasks into simpler verification steps, yielding robust feedback (Dhuliawala et al., 2023). Drawing from the extensive literature on LLM self-correction, we analyze whether VLMs can self-correct, focusing on semantic grounding.

**Prompting in LLMs and VLMs:** In-context learning in LLMs (Brown et al., 2020) has led to new prompting techniques such as Chain-of-Thought (CoT) (Wei et al., 2022), Least-to-Most (Zhou et al., 2023), and StepBack (Zheng et al., 2024) to enhance reasoning capabilities. CoT, in particular, showcases multiple reasoning paths to aid LLMs in solving complex tasks (Yao et al., 2023; Wang et al., 2023). However, these methods may be less effective in VLMs due to their limited in-context learning, especially in visually instructed VLMs (Zhao et al., 2024; Zeng et al., 2024). Conversely, zero-shot CoT promotes model reasoning without the reliance on in-context learning by simply adding a guiding sentence before model responses (Kojima et al., 2022). For VLMs, prompting has predominantly involved visual cues. Studies have shown that models, when trained on extensive web data, can recognize specific visual markers, like red circles (Shtedritski et al., 2023). More recently, Set-of-Marks (SoM) prompting has enabled the GPT-4V to ground multiple objects by overlaying object identifiers on images (Yang et al., 2023a; Nasiriany et al., 2024a). Our work incorporates these techniques to provide semantic grounding feedback to VLMs.

**Multimodal Evaluation and Verification:** Recent large-scale VLMs like CLIP (Radford et al., 2021) and GPT-4V (Yang et al., 2023b) have introduced a new paradigm in multimodal evaluation. For example, traditional metrics struggle to accurately evaluate image captions (Kilickaya et al., 2017; Cui et al., 2018). CLIPScore (Hessel et al., 2021) leverages web-scale VLMs to assess the similarity between images and captions, aligning evaluations more closely with human judgments. Similarly, LLMScore (Lu et al., 2023) combines an image captioner with an off-the-shelf object detector to measure alignment for text-to-image models directly. More recently, GPT-4V has been applied as an automatic evaluator for vision language tasks, such as text-to-3D generation and embodied question answering (Zhang et al., 2023b; Wu et al., 2024; Majumdar et al., 2024). Motivated by the potential of using large VLMs as evaluators, we investigate their capability to evaluate and verify *their own predictions*, marking a shift from earlier approaches that separated predictors from verifiers.

## 3 SELF-CORRECTION IN VLMS FOR SEMANTIC GROUNDING

In this section, we first define semantic grounding and introduce the adopted self-correction framework for VLMs in Sec. 3.1. We then introduce the key research questions on whether VLMs can correct their own grounding mistakes through self-correction in Sec. 3.2. Finally, we summarize the evaluation metrics, datasets, and models comprising our experiment protocol in Sec. 3.3.

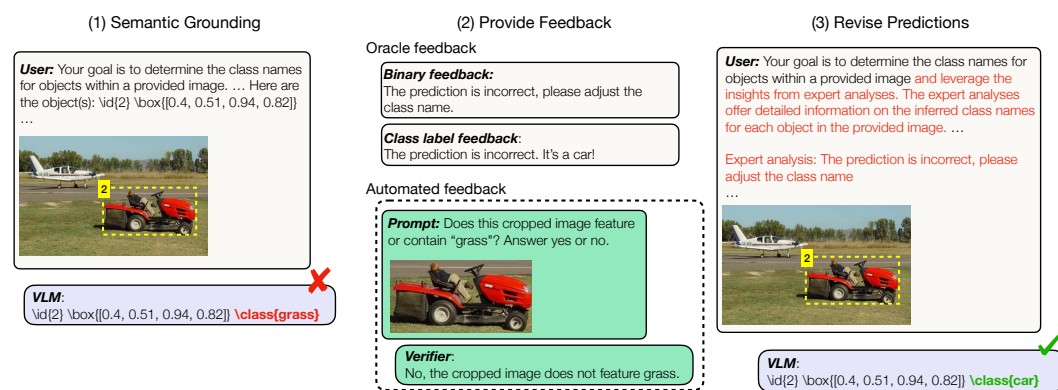

Figure 2: **Semantic grounding and self-correction framework. Left** (Semantic Grounding): Given an image and a text prompt that specifies a region of interest, a VLM is tasked to identify the semantic class best describing the image region. **Center** (Feedback Generation): For completeness, we explore both oracle and automated feedback generated from VLMs themselves. *Oracle Binary Feedback*: An oracle provides feedback only on the correctness of the predictions. *Oracle Class Label Feedback*: An oracle provides explicit feedback on the correct class labels. *Automated Binary Feedback*: A VLM acts as a 'Verifier', confirms or rejects the previous predictions. **Right** (Feedback Integration): VLMs correct their own mistakes by taking the feedback.

### 3.1 SETUP: SEMANTIC GROUNDING AND SELF-CORRECTION

**Semantic Grounding.** We study semantic grounding (Zhang et al., 2024; Yang et al., 2023a), mapping image regions to text, which Lee et al. (2024) strongly correlates with visual reasoning abilities in VLMs. Formally, consider an image $x \in \mathbb{R}^{h \times w \times 3}$ where h and w denote the image's height and width, respectively. There exists a priori image partition function that takes an image and produces N semantically distinct regions $\{r_i\}_{i=1}^{N}$, where each $r_i \in [0, 1]^{h \times w}$. A general-purpose VLM is then tasked to take the image x, the image region $r_i$, a text prompt q, and to output text $o_i = \text{VLM}(x, r_i, q)$ that best describes the image region. The output format depends on the evaluation metrics of interest. Fig 2 (left) shows an example task prompt.

Following prior works (Yang et al., 2023a; Zhang et al., 2024), we use ground truth segmentation masks as semantically distinct image regions $\{r_i\}_{i=1}^{N}$. We evaluate semantic grounding ability by whether the VLM can estimate the ground truth class label for each region in every scene.

**Self-Correction.** The term 'self-correction' are broadly adopted in LLMs (Kamoi et al., 2024). In this paper, we explore the setup involving explicit feedback generation from the VLMs. Namely, we use a 'Verifier' instantiated from the same VLM to provide feedback on the previous predictions. If feedback suggests further refinement, the VLMs then take the feedback to refine their own predictions. Fig. 2 highlights the feedback dynamics between VLMs and Verifier.

For an image $x$ and an image region $r_i$, we refer the initial predictions without feedback as *base predictions* $o_{i,0}$. For completeness, we study both oracle feedback $f^*$ and self-generated feedback $f^{\text{VLM}}$. The feedback can be converted into text or visual marks to help VLMs correct their own mistakes. Please refer to Appendix D for the complete prompt templates.

### 3.2 RESEARCH QUESTIONS

Recently, LLMs have demonstrated significant improvements in performance on complex language semantic tasks such as coding and math reasoning by leveraging self-correction (Chen et al., 2024; Nathani et al., 2023; Dhuliawala et al., 2023; Kim et al., 2023). We note that VLMs can process diverse visual and text inputs while simultaneously sustaining a dialogue from multiple input rounds similar to LLMs. To explore whether VLMs behave similarly to LLMs in self-correcting their errors in semantic grounding, we break it into two research questions **(Q1)** can VLMs receive and understand oracle feedback to improve semantic grounding? and **(Q2)** can VLMs provide high-quality binary feedback for themselves? We study binary feedback due to its lower task complexity, leading to a

**User:** Your goal is to determine the class names for objects within a provided image and leverage the insights from expert analyses. …

Expert analysis: The prediction is incorrect, please adjust the class name
…
**VLM:** After examining the image and the expert analysis

**VLM:** [VLM generates to complete]

(1) Textual prompt: Zero-shot CoT

(2) Visual prompt

Figure 3: **Examples of prompting techniques.** Left: Zero-shot CoT prepends a guiding sentence (in red) before VLMs' output. Right: We apply various visual prompting techniques including RoI crop, visual marks, and SoM to modify input images to VLMs to guide the models' attention.

more reliable feedback signal for self-correction. By systematically analyzing these two questions, we pave the way to improve semantic grounding in VLMs through self-correction *without the access to oracle feedback* in Section 4.2.

For the rest of this section, we elaborate the questions and setups.

### 3.2.1 CAN VLMs RECEIVE AND UNDERSTAND ORACLE GROUNDING FEEDBACK?

We start by asking if VLMs can receive and understand oracle grounding feedback $f^*$ to improve the base predictions. Although it is an unrealistic setup, it provides us an upper bound to improve semantic grounding in VLMs through self-correction. We study this question from two aspects: the types of feedback and the ways to prompt feedback to VLMs.

**Feedback types.** We ask: what type of feedback yields the best improvements in grounding performance? We consider two alternatives: **(i)** class label feedback – directly providing the ground truth class labels in a text prompt; and **(ii)** binary feedback – providing a message on whether the previous prediction is correct. Fig. 2 (center) visualizes the two feedback types.

**Ways to prompt feedback to VLMs.** We ask: how should the feedback be prompted to a VLM? We consider several alternatives and visualize them in Fig. 3: **(i)** *Zero-shot Chain-of-Thought (CoT)*: Motivated by Kojima et al. (2022) that shows that simply prepending a guiding sentence *'Let's think step-by-step'* before generation can strongly guide the LLMs for desired tasks, we use the guiding sentence *'After examining the image and the expert analyses, the final answer is [output_template]'* for the semantic grounding tasks. Here, the feedback is referred as expert analyses to encourage the model to follow the feedback. **(ii)** *Visual Marks*: Shtedritski et al. (2023) shows that Internet-scale vision-language encoders are biased to attend to visual marks (*e.g.*, red circles). **(iii)** *Set-of-Mark (SoM)*: Yang et al. (2023a) shows that overlaying object identifiers on the image improves visual grounding.

### 3.2.2 CAN VLMs GIVE BINARY GROUNDING FEEDBACK FOR THEMSELVES?

Prior works in LLMs suggest that feedback generation is the bottleneck in self-correction (Gou et al., 2024; Olausson et al., 2024). The survey paper in LLMs (Kamoi et al., 2024) identifies that decomposing complex generation tasks into easier verification tasks enables successful self-correction (Dhuliawala et al., 2023). Following this insight, we study binary feedback, a message on whether the previous prediction is correct. We refer the VLMs performing verification to as 'Verifier'. We study binary feedback verification by comparing it with generation-based verification (Madaan et al., 2023; Kim et al., 2023) referred to as "intrinsic self-correction" in prior work (Huang et al., 2023). Furthermore, we also study the proper ways to prompt the Verifier.

**Baseline approach: intrinsic self-correction.** We adopt prior work in LLMs self-correction (Kim et al., 2023) to semantic grounding task. Here, we prompt the verifier to *'Carefully review and refine your answer'* right after the base predictions to automatically correct grounding predictions. Although

|  | Zero-shot CoT | Visual Prompt | LLaVA-1.5 | ViP-LLaVA | CogVLM |
|---|---|---|---|---|---|
| Base Predictions | N/A | No | 35.86 | 35.86 | 15.98 |
| + Class Label Feedback | No | No | $94.80_{+58.94}$ | $74.99_{+39.13}$ | $77.04_{+61.06}$ |
| + Binary Feedback | No | No | 41.04 | 40.36 | 16.25 |
|  | Yes | No | 43.30 | 42.00 | 18.25 |
|  | Yes | SoM | 42.41 | 44.53 | 18.64 |
|  | Yes | Visual marks | $\mathbf{45.38}_{+9.52}$ | $\mathbf{45.21}_{+9.35}$ | $\mathbf{19.46}_{+3.48}$ |

Table 1: **VLMs use oracle feedback to improve grounding accuracy.** We explore how oracle Class Label Feedback and Binary Feedback improve semantic grounding in VLMs. For each type of feedback and VLM, we highlight the largest improvements w.r.t. the performance of its base predictions.

intrinsic self-correction doesn't explicitly generate binary feedback, a binary signal can be obtained by comparing the alignment of grounding predictions before and after correction

**Ways to prompt the Verifier.** We consider several techniques and visualize them in Fig 3: **(i)** *Visual marks:* The verifier receives the image with a highlighted object of interest and a prompt to determine if the predicted class label accurately describes the object (Shtedritski et al., 2023). **(ii)** *RoI crop:* Prior work (Gu et al., 2022) distills features of cropped regions to the object detectors. Inspired by this, we design the verifier to receive a cropped image isolating the object of interest. **(iii)** A combination of Visual Marks and RoI crop.

## 3.3 EXPERIMENT PROTOCOLS

**Datasets.** We analyze the panoptic segmentation dataset from ADE20k (Zhou et al., 2017), which was not previously used for instruction tuning in the open-source VLMs under study. This dataset includes a validation set comprising 2k complex, crowded scenes with over 30k masks across 150 distinct categories. We further validate our results in the iterative setting of COCO panoptic segmentation (Kirillov et al., 2019; Lin et al., 2014). Although the COCO dataset is a standard in visual grounding, most VLMs train on a visual instruction dataset derived from COCO, making it in-domain, unlike ADE20k. The COCO validation set consists of 5k images. Consistent with previous VLM grounding research (Yang et al., 2023a), we utilize the same subset of 100 images from both ADE20k and COCO for our analysis.

**VLMs.** We analyze three state-of-the-art open-source VLMs including LLaVA-1.5 (Liu et al., 2023a), ViP-LLaVA (Cai et al., 2024) and CogVLM (Wang et al., 2024). LLaVA-1.5 is a successor of LLaVA (Liu et al., 2023b), a visual instruction tuned VLM, and has scaled up to a larger model and a larger training dataset. ViP-LLaVA shares the overall model architecture and training strategy with LLaVA, but focuses on synthesizing a diverse set of visual marks in the training dataset, effectively improving the model performance when using visual prompts and allowing for a more user-friendly interface. CogVLM is a generalist VLM with highlights on integrating image and text features without sacrificing any performance on NLP tasks.

**Grounding metrics.** We evaluate semantic grounding performance by measuring classification accuracy. We use off-the-shelf sentence embeddings (Huggingface) to map the VLM outputs $o_i$ to the label from the class label list with the largest cosine similarity. We then report accuracy aggregated over all regions $r_i$ for each scene in the dataset. While it is not idea, our quantitative analysis in Appendix B demonstrates that the errors are within a reasonable range.

**Feedback metrics.** We assess the VLM verifier's capability to generate a binary feedback signal by measuring the $F_1$ scores, considering the imbalanced distribution of oracle binary feedback. In Appendix H.1, we show that $F_1$ is a more representative metric than accuracy for evaluating feedback quality.

|  | Visual prompt | LLaVA-1.5 | ViP-LLaVA | CogVLM |
|---|---|---|---|---|
| Intrinsic Self-Correction | N/A | 51.12 | 48.19 | 21.87 |
| VLM Binary Verification | Visual marks | 56.16 | **60.47** | 39.16 |
|  | RoI crop | **61.71** | 58.18 | **40.68** |
|  | Visual marks + RoI crop | 61.14 | 59.6 | 39.79 |

Table 2: **VLM binary verification provide higher-quality binary feedback (higher $F_1$ scores) compared to intrinsic self-correction.** The choices of visual prompting techniques should be tailored to the specific VLMs. We bold the best performances of each VLM.

## 4 EMPIRICAL FINDINGS

In this section, we experiment on the ADE20k dataset to study the questions in Sec. 3.2. All experiments are run on three different seeds and we report the average performances. We release the code at here.

### 4.1 CAN VLMS RECEIVE AND UNDERSTAND ORACLE GROUNDING FEEDBACK?

Table 1 summarizes the base predictions for each model and the improved grounding accuracies after receiving oracle grounding feedback.

**Findings of feedback types.** We first compare the improvement in accuracy with no additional prompting techniques (*i.e.*, zero-shot CoT or visual prompts). Table 1 shows that oracle class label and binary feedback improve grounding accuracy by up to 61.06 and 5.18, respectively. We find that VLMs can receive and understand oracle feedback to improve performance, without requiring any additional data, training time, or architectural modifications.

Intuitively, oracle class label feedback yields the most improvement, since it directly reveals the class labels and consequently reduces the semantic grounding task to a text retrieval problem. Perhaps surprisingly, oracle class label feedback does not automatically improve accuracy to 100%. This outcome highlights a limitation in open-source VLMs' ability to perform tasks based solely on language understanding, indicating a potential area for improvement in these models (Lin et al., 2023). Indeed, some models fail in approximately 25% of cases in this scenario, demonstrating a significant deficiency in prompt-following capabilities that warrants further investigation. (see Table 1, Class Label Feedback)

**Findings of ways to prompt feedback to VLMs.** Table 1 shows that zero-shot CoT augments oracle binary feedback for every model considered by up to 2.26 accuracy points. This aligns with trends in LLMs that suggest the effectiveness of CoT to improve reasoning (Wei et al., 2022; Kojima et al., 2022). On the other hand, visual prompting with SoM (Yang et al., 2023a) does not significantly improve beyond zero-shot CoT for models that were not already pre-trained with data featuring visual prompting cues (*e.g.*, LLaVa-1.5). In contrast, ViP-LLaVA was specifically trained for interpreting visual cues; this model improves with both SoM and visual marks (*e.g.*, red circles). Notably, the combination of zero-shot CoT and visual marks emerges as the most effective strategy, increasing by 7.45 grounding accuracy points relative to the base predictions. Thus, for open-source VLMs, we identify that the best way to introduce binary feedback in semantic grounding is to combine visual marks and zero-shot CoT.

### 4.2 CAN VLMS GIVE BINARY GROUNDING FEEDBACK FOR THEMSELVES?

We assess the quality of binary feedback using $F_1$ scores due to potentially imbalanced oracle feedback. Table 2 provides $F_1$ scores of intrinsic self-correction and the binary feedback produced by a VLM Verifier.

**Results.** We first assess the effectiveness of intrinsic self-correction, which involves continuing another round of conversation by asking *'Carefully review and refine your answer'* to the VLM and directly outputting the revised predictions. We derive the binary feedback by comparing whether the revised predictions differ from the initial predictions. When evaluated in accuracy, intrinsic self-correction achieves low accuracies at 47.03, 47.13, and 59.5 on LLaVA-1.5, ViP-LLaVA, and

| VLM | Binary feedback source | Dialogue round | | | | | |
|---|---|---|---|---|---|---|---|
| | | $t = 0$ | $t = 1$ | $t = 2$ | $t = 3$ | $t = 4$ | $t = 5$ |
| LLaVA-1.5 | Intrinsic Self-Correction | 35.86 | 30.92 | 29.64 | $28.54_{-7.32}$ | - | - |
| | VLM Verification (**ours**) | 35.86 | 37.97 | 38.93 | 39.27 | 39.54 | $40.29_{+4.43}$ |
| | Oracle Verification (**ours**) | 35.86 | 45.42 | 47.95 | 51.55 | 52.04 | $53.2_{+17.34}$ |
| ViP-LLaVA | Intrinsic Self-Correction | 35.86 | 27.72 | 26.7 | $25.68_{-10.18}$ | - | - |
| | VLM Verification (**ours**) | 35.86 | 35.14 | 36.06 | 36.37 | 36.16 | $36.47_{+0.39}$ |
| | Oracle Verification (**ours**) | 35.86 | 47.45 | 47.64 | 50.54 | 51.82 | $53.13_{+17.27}$ |
| CogVLM | Intrinsic Self-Correction | 15.98 | 8.33 | 8.6 | $9.08_{-6.9}$ | - | - |
| | VLM Verification (**ours**) | 15.98 | 17.13 | 17.96 | 18.09 | 18.5 | $18.64_{+2.66}$ |
| | Oracle Verification (**ours**) | 15.98 | 19.6 | 20.96 | 21.51 | 21.82 | $22.12_{+6.14}$ |
| GPT-4V | Intrinsic Self-Correction | 40.36 | 22.33 | 25.2 | $22.95_{-17.41}$ | - | - |
| | VLM Verification (**ours**) | 40.36 | 41.8 | 43.23 | $42.4_{+2.04}$ | - | - |
| | Oracle Verification (**ours**) | 40.36 | 50 | 52.45 | $53.27_{+12.91}$ | - | - |
| GPT-4o | Intrinsic Self-Correction | 33.81 | 34.01 | 39.13 | $37.5_{+3.68}$ | - | - |
| | VLM Verification (**ours**) | 33.81 | 39.13 | 40.98 | $41.18_{+7.36}$ | - | - |
| | Oracle Verification (**ours**) | 33.81 | 49.59 | 54.91 | $57.78_{+23.91}$ | - | - |

Table 3: **Iterative VLM binary feedback improves grounding accuracy in ADE20k.** We highlight the performance difference w.r.t. the performance of the base predictions and if the performances are below the performance of the base predictions, we use red-colored font.

CogVLM, respectively. The VLMs results here are aligned with previous studies on LLMs Kamoi et al. (2024) that LLMs struggle to improve via intrinsic self-correction out-of-the-box.

In Table 2, we identify that binary verification mechanism for VLM using RoI crop significantly improves the $F_1$ score for all three models, by up to 18.81 points. This observation aligns well with the strong self-evaluation capabilities in LLMs. We may also augment this binary verification with visual marks such as red circles. Additionally, the choice of visual prompting technique should be tailored to the specific VLM. For instance, RoI crop tends to be more effective for networks not trained on visual marks (*e.g.*, LLaVA-1.5 and CogVLM), while visual marks yield better results for models accustomed to such cues (*e.g.*, ViP-LLaVA).

# 5 CAN VLMs CORRECT THEIR GROUNDING ERRORS THROUGH SELF-CORRECTION?

Our key findings in Sec. 4 show that **(1)** VLMs can receive and understand oracle feedback and **(2)** VLMs can given binary feedback for themselves. We now combine them to evaluate whether VLMs can self-correct their mistakes by leveraging another instance of the same model. Furthermore, can VLMs *iteratively* perform self-correction to trade compute for performances?

## 5.1 SETUP: ITERATIVE SELF-CORRECTION IN VLMs

We introduce an iterative dialogue loop between a VLM agent and Verifier, where at the first timestep $t = 0$, the VLM obtains base predictions $\{o_{i,0}\}_{i=1}^{N}$ for every scene (Sec. 3.1). We then prompt the Verifier to generate a binary feedback signal for every prediction $f^{\text{VLM}}(x, r_i, o_{i,0})$ (Sec. 3.2.2). In the next timestep, the VLM agent uses this binary feedback to revise predictions (Sec. 3.2.1). We repeat these steps to a maximum iteration count or until the verifier agrees with the prediction.

In our experiments, we use the textual prompts (*i.e.*, zero-shot CoT) and the visual prompts (*i.e.*, red circles for open-source VLMs and SoM for proprietary VLMs) to encourage feedback receiving and use RoI crop when VLMs provide binary feedback. Consistent with prior work (Yang et al., 2023a), we use the same subset of 100 images for ADE20k and COCO for our analysis.

**Baselines.** We adopt the same baseline used in Sec. 3.2.2: intrinsic self-correction adopted from prior work in LLMs (Kim et al., 2023). To identify the self-correction upper bounds of each VLM,

we also report the performances of self-correction with the access to oracle binary feedback, referred to as Oracle Verification.

**Proprietary VLMs.** Open-source VLMs often suffer from shorter context window or limited instruction following capabilities. We, therefore, experiment the identified self-correction framework using GPT-4V (Yang et al., 2023b) and its successor GPT-4o.

**Base predictions generation.** The self-correction survey in LLMs (Kamoi et al., 2024) finds that the weak initial predictions can lead to false promises in self-correction. We attempted to improve open-source VLMs by adding SoM prompt, but observed significant performance drops compared to using bounding boxes alone. For LLaVA-1.5, the base predictions achieve 35.86 in ADE20k. However, adding SoM and using RoI crop result in 11.06 and 19.67, respectively. This may be because most open-source VLMs, including the three in our study, are trained to identify image regions using bounding boxes (Zhang et al., 2024; You et al., 2023). In contrast, proprietary VLMs have shown strong improvements with SoM (Yang et al., 2023a). Therefore, we adopted SoM to generate base predictions for GPT-4V and GPT-4o.

### 5.2 MAIN RESULTS

**Open-source VLMs.** Tables 3 and 4 illustrate that multiple rounds of oracle binary feedback consistently enhance the performance of all open-source VLMs, with gains ranging from 6.14 to 17.34 in ADE20k and 6.45 to 15.28 in COCO. Additionally, multiple self-correction increase grounding accuracy by up to 7.78 and 7.64 points on ADE20k and COCO, respectively, compared to a single round (*i.e.*, t = 1). The identified VLM binary verification, despite producing noisy feedback, also *consistently* improves grounding accuracy by 0.39 to 4.43 points in ADE20k and 1.91 to 4.04 points in COCO. These gains are consistent across all three open-source VLMs, underscoring the benefits of iterative feedback for zero-shot improvements in grounding accuracy, even with noisy feedback.

In sharp contrast, intrinsic self-correction decreases downstream grounding in all settings by up to 10 points, except where base predictions are weak, such as with CogVLM in COCO. We empirically find that self-correction cannot reliably identify the already correct predictions.

**GPT-4V and GPT-4o.** GPT-4V and GPT-4o improve predictions with both VLM binary feedback and oracle binary feedback, even more than the open-source VLMs do. In particular, GPT-4o significantly improves and sometimes surpasses GPT-4V, especially when incorporating oracle binary feedback. However, perhaps surprising, even with oracle binary feedback indicating prediction correctness, strong GPT-4V and GPT4o fail to provide correct responses after three turns with less than 60 points accuracy overall in ADE20k.

Similar to open-source VLMs, GPT-4V exhibit negative improvements in intrinsic self-correction. Intriguingly, there are stark differences between GPT-4V and GPT-4o: GPT-4V shows a 17-point decrease in accuracy in ADE20k, while GPT-4o sees a 7-point increase in COCO. The reasons for these sharp differences remain unclear due to unknown model architectures and specific training data used in proprietary models. However, we note that the identified VLM binary verification consistently improves upon both base predictions and intrinsic self-correction with enough dialogue rounds.

We emphasize that the identified VLM binary feedback verification requires no access to external tool or oracle. Thus, our results show that *VLMs can iteratively self-correct their own grounding mistakes when prompted in a proper way.* We anticipate the improvements from iterative self-correction will improve with future VLMs.

## 6 CONCLUSION

In this work, we explore the potentials of self-correction in large vision-language models in the context of semantic grounding. We break this research question by asking two key questions **(Q1)** Can VLMs receive and understand oracle grounding feedback and **(Q2)** Can VLMs provide grounding feedback? Throughout our systematic analysis, we find that the answers to both questions are positive when prompted in a proper way. With two datasets and five VLMs including proprietary ones, we demonstrate that with the identified VLM binary feedback verification, VLMs *can* iterative self-correct their own grounding mistakes. Within five rounds of VLM binary feedback, open-

| VLM | Binary feedback source | Dialogue round | | | | | |
|---|---|---|---|---|---|---|---|
| | | $t = 0$ | $t = 1$ | $t = 2$ | $t = 3$ | $t = 4$ | $t = 5$ |
| LLaVA-1.5 | Intrinsic Self-Correction | 36.3 | 33.69 | 32.26 | $31.63_{-4.66}$ | - | - |
| | VLM Verification (**ours**) | 36.3 | 35.87 | 36.94 | 37.04 | 37.69 | $38.21_{+1.91}$ |
| | Oracle Verification (**ours**) | 36.3 | 41.55 | 43.81 | 46.22 | 47.55 | $48.77_{+12.47}$ |
| ViP-LLaVA | Intrinsic Self-Correction | 37.26 | 32.64 | 32.4 | $31.12_{-6.13}$ | - | - |
| | VLM Verification (**ours**) | 37.26 | 37.84 | 39.64 | 39.64 | 40.01 | $40.44_{+3.18}$ |
| | Oracle Verification (**ours**) | 37.26 | 44.9 | 48.08 | 50.15 | 51.75 | $52.54_{+15.28}$ |
| CogVLM | Intrinsic Self-Correction | 14.8 | 16.23 | 16.47 | $15.92_{+1.11}$ | - | - |
| | VLM Verification (**ours**) | 14.8 | 16.97 | 17.83 | 18.3 | 18.52 | $18.84_{+4.04}$ |
| | Oracle Verification (**ours**) | 14.8 | 19.42 | 20.14 | 20.7 | 21.01 | $21.25_{+6.45}$ |
| GPT-4V | Intrinsic Self-Correction | 40.92 | 30.89 | 36.62 | $32.8_{-8.12}$ | - | - |
| | VLM Verification (**ours**) | 40.92 | 43.94 | 44.9 | $45.38_{+4.46}$ | - | - |
| | Oracle Verification (**ours**) | 40.92 | 52.7 | 56.5 | $57.8_{+16.88}$ | - | - |
| GPT-4o | Intrinsic Self-Correction | 39.49 | 47.13 | 48.08 | $46.65_{+7.15}$ | - | - |
| | VLM Verification (**ours**) | 39.49 | 46.49 | 47.77 | $47.92_{+8.43}$ | - | - |
| | Oracle Verification (**ours**) | 39.49 | 57 | 62.26 | $67.19_{+27.69}$ | - | - |

Table 4: **Iterative VLM binary feedback improves grounding accuracy in COCO.** We highlight the performance difference w.r.t. the performance of the base predictions and if the performances are below the performance of the base predictions, we use red-colored font.

source VLMs and proprietary VLMs improve up to 4 and 8 accuracy points. We highlight that the self-correction in VLMs requires no access to oracle or any finetuning or architectural changes.

**Limitations.** Despite the advances in VLMs' semantic grounding through self-correction, this approach trades compute for performance. Appendix C shows the GPT-4o performance-cost tradeoff. Therefore, in applications requiring low latency, feedback-based reasoning becomes less practical. Additionally, assessing VLMs' zero-shot capabilities with close-set vocabularies highlights language ambiguities. For instance, in ADE20k, similar classes like 'grass', 'field', 'plant', and 'tree' exacerbate this issue. For proprietary VLMs, we include the class list in the prompt, but this does not resolve ambiguities as each dataset may interpret classes differently. For open-source VLMs, given the smaller context window, we rely on off-the-shelf embeddings for mapping, which can introduce noise. We provide additional quantitative analysis on the errors in class mapping in Appendix B. We expect future generations of open-source VLMs to achieve significant quantitative improvements in these tasks.

**Ethics Statement.** This paper discusses self-correction in VLMs. The identified self-correction framework promotes a cost-effective way to improve semantic grounding in VLMs and allow continuous refinement with minimal resources, *i.e.* require no further fine-tuning. However, the abilities to take noisy feedback might bring further risks to VLMs with a long context window if the multiple adversarial feedback are provided as in-context examples, similar to the risks raised in Anil et al.

**Reproducibility Statement.** We provide the full prompt in Appendix D and detailed implementation in Appendix G. The sampled dataset can be access in the official github repository in prior work (Yang et al., 2023a). We release the code at here.

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
