## A    TABLE OF CONTENTS

## B    QUANTITATIVE ANALYSIS IN CLASS MAPPING ERRORS

Assessing VLMs' zero-shot capabilities with close-set vocabularies highlights language ambiguities. In this work, we rely on off-the-shelf sentence embeddings for the class mapping. To quantify errors introduced by mapping model outputs to close-set class labels, we conducted an additional experiment: We sampled 100 raw outputs from LLaVA-1.5 in ADE. A human (one of the authors) evaluated whether the mapping from raw output to class labels, using sentence embeddings, was correct. Table 5 Evaluating open-vocabulary models cheaply and automatically remains an open question. Even human evaluators found 10% of the data difficult to map correctly. We have tried to ensure fair comparisons between approaches by maintaining consistent mapping.

| Options | Counts |
|---|---|
| The mapping is correct. | 77 |
| The mapping is incorrect and I can provide the correct one. | 13 |
| The mapping is incorrect, but it is hard to find a good one from the close set class labels. | 10 |
| Total | 100 |

Table 5: **Human studies in quantifying the error in class mapping.**

## C    PERFORMANCE-COST TRADEOFF

Despite the advances in VLMs' semantic grounding through self-correction, the identified self-correction trades compute for performance. Fig. 4 shows the GPT-4o performance-cost tradeoff in ADE20k.

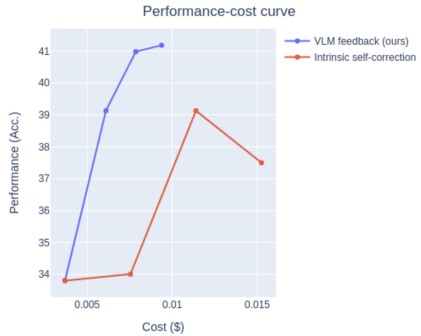

Figure 4: **Cost-performance tradeoff of GPT-4o in ADE20k**

# D PROMPT TEMPLATES

We show the full prompt templates

1. To producing base semantic grounding predictions in Fig. 5

2. To enhance previous semantic grounding predictions by taking binary feedback in Fig. 6

3. To enhance previous semantic grounding predictions by taking class label feedback in Fig. 7

4. To produce VLM binary feedback in Fig. 8.

5. For the GPT-4V and GPT-4o experiments, we provide the class names by appending '*You must answer by selecting from the following names: [COCO or ADE20k Vocabulary]*' in the prompt[1], as shown in Fig. 9 and Fig. 10.

```
User: You are tasked with visual semantic grounding. Your
↪   goal is to determine the class names for objects within a
↪   provided image. Each object in the image is identified by
↪   a unique ID and its location is defined by a precise
↪   bounding box, formatted as: \id{id} \box{[x1, y1, x2,
↪   y2]}, where coordinates specify the box corners. The
↪   inferred class name for each object is denoted as
↪   \class{class name}. Here are the objects: \id{2}
↪   \box{[0.1, 0.2, 0.13, 0.43]}
Put your final answer by filling in the placeholder(s) in the
↪   following string at the beginning: "\id{2} \box{[0.1,
↪   0.2, 0.13, 0.43]} \class{your answer here}"
```

Figure 5: Prompt template to produce the base predictions. The text in red represents variables.

# E EXAMPLE DIALOGUE

In Fig. 11, we demonstrate the iterative interactions between a VLM agent and the Verifier. In Fig. 12, we show the effectiveness of VLM binary verification in GPT-4V [2].

# F DATASET DETAILS

We use ADE20k and COCO panoptic segmentation dataset to evaluate the semantic grounding performance in VLMs. We adopt SoM split provided in the prior work Yang et al. (2023a)[3]. ADE20k is a large-scale dataset with fine-grained segmentation labels. We adopt the variant with 150 classes, commonly referred to as ADE20k-150. COCO panoptic segmentation is a standard dataset to evaluate visual grounding. There are 133 fine-grained classes in total, composed of 80 thing classes and 53 stuff classes. Consistent with prior works, SoM (Yang et al., 2023a), we use the same subset of 100 images for ADE20k and COCO for our analysis. There are 100 images and 488 segmentation masks in ADE20k SoM split and 101 and 628 segmentation masks in COCO SoM split.

Every region $r_i$ in ADE20k and COCO panoptic segmentation dataset is represented with segmentation mask. We convert them to a more compact representation, *i.e.* bounding box, and feed them to the VLMs in the text prompt

---

[1] https://github.com/microsoft/SoM/tree/main/benchmark#open-vocab-segmentation-on-coco

[2] GPT-4V predictions with simplified prompts as of Mar 22, 2024: https://imgur.com/a/nbKjIlb

[3] https://github.com/microsoft/SoM/tree/main/benchmark#dataset

```
User: You are tasked with visual semantic grounding. Your
    ↪  goal is to determine the class names for objects within a
    ↪  provided image and leverage the insights from expert
    ↪  analyses. The expert analyses offer detailed information
    ↪  on the inferred class names for each object in the
    ↪  provided image. Each object in the image is identified by
    ↪  a unique ID and its location is defined by a precise
    ↪  bounding box, formatted as: \id{id} \box{[x1, y1, x2,
    ↪  y2]}, where coordinates specify the box corners. The
    ↪  inferred class name for each object is denoted as
    ↪  \class{class name}. I have labeled each object with its
    ↪  ID and overlaid its segmentation mask on the image to
    ↪  clarify the correspondences.

One expert analyses on the provided image are shown below:
* Analysis 1
Object(s) with inferred class names: \id{2} \box{[0.1, 0.2,
    ↪  0.13, 0.43]} \class{wall}
Expert's decision(s) on class names: The inferred class
    ↪  name(s) for {incorrect obj id} are incorrect. The
    ↪  inferred class name(s) for \id{2} are not "wall".
Expert's suggestion: Adjust the class names for objects with
    ↪  IDs \id{2}

Examine the image and the expert analyses to determine the
    ↪  true class name of the object(s): \id{2} \box{[0.1, 0.2,
    ↪  0.13, 0.43]}. Put your final answer by filling in the
    ↪  placeholder(s) in the following string at the beginning:
    ↪  "\id{2} \box{[0.1, 0.2, 0.13, 0.43]} \class{your answer
    ↪  here}"
```

Figure 6: Prompt template to improve semantic grounding predictions by taking Binary Feedback. The text in red represents variables.

# G    IMPLEMENTATION DETAILS

Every experiment throughout this paper is run over three seeds and we report the average scores except for experiments with proprietary VLMs. All the experiments are run in a single-node machine with two A40 GPUs. In the experiments with binary or class label feedback, we only ask VLMs to correct those that are incorrect based on the feedback. Therefore, if the feedback is noisy, *e.g.* VLM binary verification, VLMs can possibly decrease the performances. See Fig. 16 for example.

**Open-source VLMs.** We adopt LLaVA-1.5 13b (from https://huggingface.co/llava-hf/llava-1.5-13b-hf), ViP-LLaVA 13b (from https://huggingface.co/llava-hf/vip-llava-13b-hf), and CogVLM (from https://huggingface.co/THUDM/CogVLM). When perform the VLM forward pass $o_i = \text{VLM}(x, r_i, q)$, we set the temperature to 0.9, top_p to 0.8, max_new_tokens to 1024, and draw five samples per forward pass. We take the majority vote responses as the final answers $o_i$.

**GPT-4V.** As suggested in prior work (Yang et al., 2023a,b), GPT-4V exhibits better grounding ability when the objects are specified by visual prompts rather than text prompts. Therefore, we adopt GPT-4V & SoM to obtain the base predictions, where we overlay object masks and numeric identifiers on the images. Furthermore, when using VLMs to produce feedback, we apply SoM to specify each object. Finally, since GPT-4V has a longer context window compared to open-source VLMs, we include the class list in the prompt to encourage better alignment between the responses and the ground truth. All GPT-4V experiments are done over the OpenAI API and we follow

```
User: You are tasked with visual semantic grounding. Your
↪  goal is to determine the class names for objects within a
↪  provided image and leverage the insights from expert
↪  analyses. The expert analyses offer detailed information
↪  on the inferred class names for each object in the
↪  provided image. Each object in the image is identified by
↪  a unique ID and its location is defined by a precise
↪  bounding box, formatted as: \id{id} \box{[x1, y1, x2,
↪  y2]}, where coordinates specify the box corners. The
↪  inferred class name for each object is denoted as
↪  \class{class name}. I have labeled each object with its
↪  ID and overlaid its segmentation mask on the image to
↪  clarify the correspondences.

One expert analyses on the provided image are shown below:
* Analysis 1
Object(s) with inferred class names: \id{2} \box{[0.1, 0.2,
↪  0.13, 0.43]} \class{wall}
Expert's decision(s) on class names: The inferred class
↪  name(s) for \id{2} are incorrect. The inferred class
↪  name(s) for \id{2} are not "wall".
Expert's suggestion: Adjust the class names for objects with
↪  IDs \id{2} to \class{rail}.

Examine the image and the expert analyses to determine the
↪  true class name of the object(s): \id{2} \box{[0.1, 0.2,
↪  0.13, 0.43]}. Put your final answer by filling in the
↪  placeholder(s) in the following string at the beginning:
↪  "\id{2} \box{[0.1, 0.2, 0.13, 0.43]} \class{your answer
↪  here}"
```

Figure 7: Prompt template to improve semantic grounding predictions by taking Class Label Feedback. The text in red represents variables.

```
User: Does this cropped image contain "wall"? Answer yes or
↪  no.
```

Figure 8: Prompt template to derive VLM binary feedback. The text in red represents variables.

the exact same evaluation procedures described in Sec. 3.3, where we use the off-the-shelf text embeddings (Huggingface) to map the GPT-4V outputs $o_i$ to the nearest label from the class label list.

We follow the implementation provided in Yang et al. (2023a)[4] and set the system prompt as: - *For any marks mentioned in your answer, please highlight them with [].* We follow Yang et al. (2023a) to set the alpha parameters in SoM as 0.2 and 0.4 in ADE20k and COCO, respectively. We use the endpoint gpt-4-0125-preview.

**GPT-4o.** Similar to GPT-4V, we empirically found that SoM prompts improve the base predictions in the semantic grounding tasks in ADE20k. We, therefore, hypothesize that GPT-4o benefits by having SoM prompts. We use the endpoint gpt-4o-2024-05-13.

---

[4]https://github.com/microsoft/SoM/blob/main/gpt4v.py

```
User: I have labeled a bright numeric ID at the center for
↪   each visual object in the image. Please enumerate their
↪   names. You must answer by selecting from the following
↪   names: [Class list]
```

Figure 9: Prompt template for GPT-4V and GPT-4o to produce the base predictions. Following prior work (Yang et al., 2023a), we include the full class list in the text prompt. The text in red represents variables.

```
User: You are tasked with visual semantic grounding. Your
↪   goal is to determine the class names for objects within a
↪   provided image and leverage the insights from expert
↪   analyses. The expert analyses offer detailed information
↪   on the inferred class names for each object in the
↪   provided image. Each object in the image is identified by
↪   a unique ID and its location is defined by a precise
↪   bounding box, formatted as: \id{id} \box{[x1, y1, x2,
↪   y2]}, where coordinates specify the box corners. The
↪   inferred class name for each object is denoted as
↪   \class{class name}. I have labeled each object with its
↪   ID and overlaid its segmentation mask on the image to
↪   clarify the correspondences.

One expert analyses on the provided image are shown below:
* Analysis 1
Object(s) with inferred class names: \id{2} \box{[0.1, 0.2,
↪   0.13, 0.43]} \class{wall}
Expert's decision(s) on class names: The inferred class
↪   name(s) for {incorrect obj id} are incorrect. The
↪   inferred class name(s) for \id{2} are not "wall".
Expert's suggestion: Adjust the class names for objects with
↪   IDs \id{2}

Examine the image and the expert analyses to determine the
↪   true class name of the object(s): \id{2} \box{[0.1, 0.2,
↪   0.13, 0.43]}. Put your final answer by filling in the
↪   placeholder(s) in the following string at the beginning:
↪   "\id{2} \box{[0.1, 0.2, 0.13, 0.43]} \class{your answer
↪   here}"

You must answer by selecting from the following names: [ADE
↪   Class List]
```

Figure 10: Prompt template for GPT-4V to improve semantic grounding predictions by taking Binary Feedback. Following prior work Yang et al. (2023a), we include the full class list in the text prompt. The text in red represents variables.

# H ADDITIONAL RESULTS

## H.1 FEEDBACK ACCURACY DOES NOT STRONGLY CORRELATE WITH SEMANTIC GROUNDING WITH ITERATIVELY SELF-GENERATED FEEDBACK

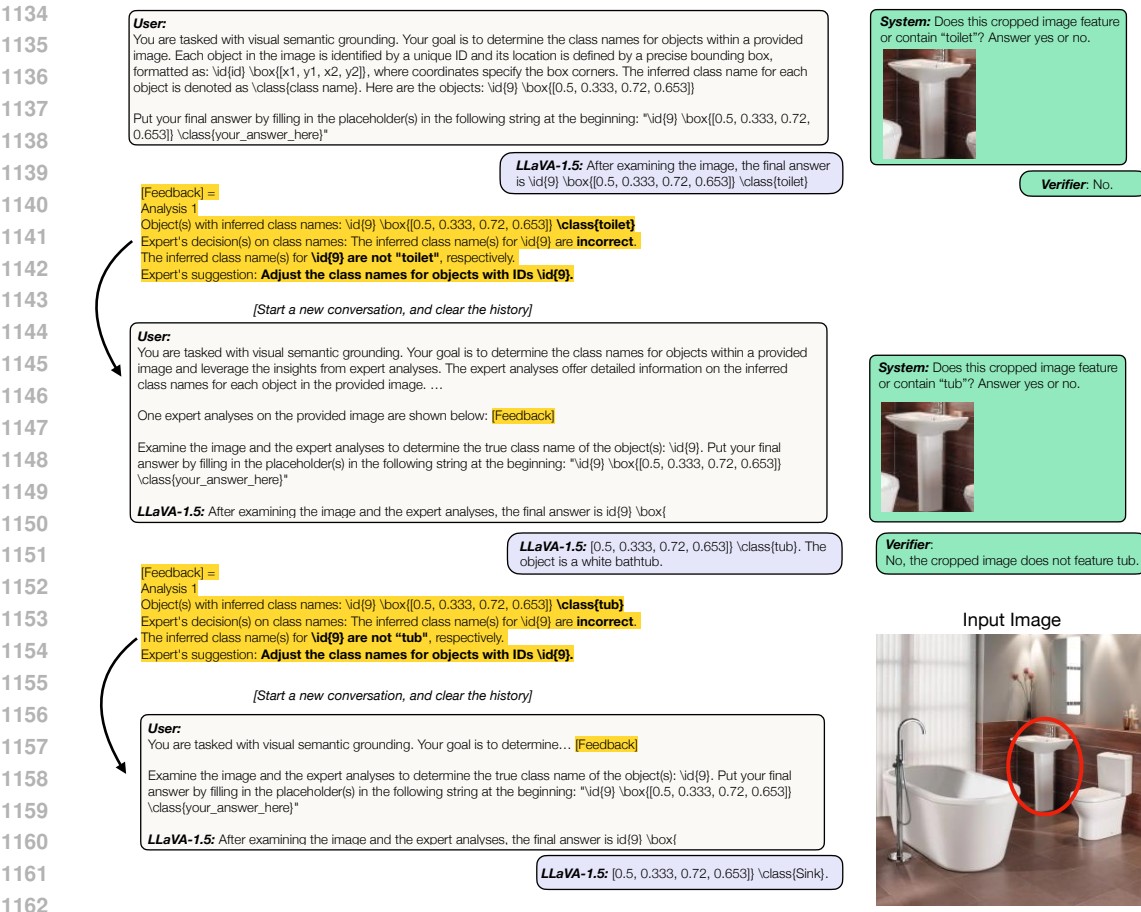

Figure 11: **Example dialogue of using iterative binary self-feedback to improve semantic grounding in VLMs.** Initially, the user queries the semantic class of an object within a bounding box. At the first attempt, the VLM responds without feedback. To refine accuracy, we use the verifier (the same VLM) to answer a yes/no question for binary self-feedback. Incorporating this feedback, we prompt the VLM again, leading to a refined prediction. The VLM's initial guess evolves from 'toilet' to 'bathtub', and ultimately to 'sink' – the correct classification.

| | Visual prompt | LLaVA-1.5 | ViP-LLaVA | CogVLM |
|---|---|---|---|---|
| Intrinsic Self-Correction | N/A | 47.03 | 47.13 | **59.5** |
| VLM Binary Verification | Visual marks | 55.5 | 65.2 | 52.3 |
| | RoI crop | **64.1** | 57.6 | 57 |
| | Visual marks + RoI crop | 62.1 | **67.2** | 52.9 |

Table 6: **Accuracy of the VLMs binary feedback** $Acc_{\text{feedback}}$. We find that intrinsic self-correction often improves accuracy in VLMs with lower base prediction performance due to imbalanced oracle binary feedback.

In the main paper, we measure feedback in $F_1$ score. Another intuitive evaluation metric is feedback accuracy, denoted as $Acc_{\text{feedback}}$ and we show the results in Table 6. We find that VLM binary verification with a higher $Acc_{\text{feedback}}$ *does not necessary* lead to a higher grounding accuracy in the iterative setup in Sec. 4.2. On average, we find that $Acc_{\text{feedback}}$ achieve an 0.11 Spearman rank correlation coefficient Spearman (1987) with grounding accuracy at t = 3 as compared to 0.72 achieved by $F_1$. We conclude that $F_1$ is a better evaluation metric for measure feedback quality in this work.

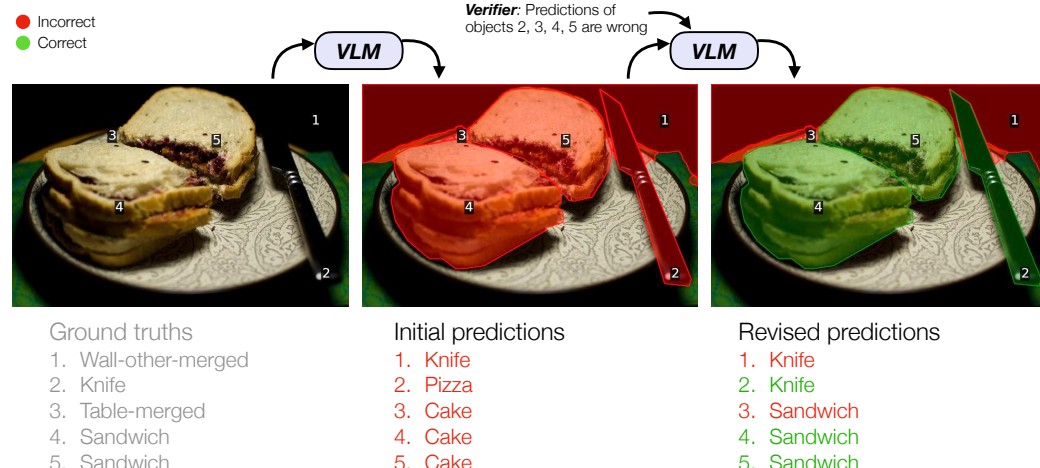

Figure 12: **Enhancing semantic grounding in VLMs with self-generated feedback.** We use GPT-4V as the VLM here. From the left to the center figure, GPT-4V takes the SoM-prompted image Yang et al. (2023a) as input and struggles to predicts the class names of each object. From the center to the right figure, GPT-4V takes the same SoM-prompted image and the additional feedback from the verifier and successfully correct the class names of three out of five objects. The verifier is another GPT-4V that operates on an altered input image and may produce noisy feedback, *e.g.*, misclassify object 1 as correct.

## H.2 QUALITATIVE RESULTS

We share additional qualitative results on ADE20k and COCO in Fig. 13, Fig. 14, Fig. 15. We also note that most of the failure cases occur when 1) the VLMs keep their own predictions even though the feedback refers them as incorrect predictions or 2) when the self-generated feedback is incorrect, as shown in Fig. 16.

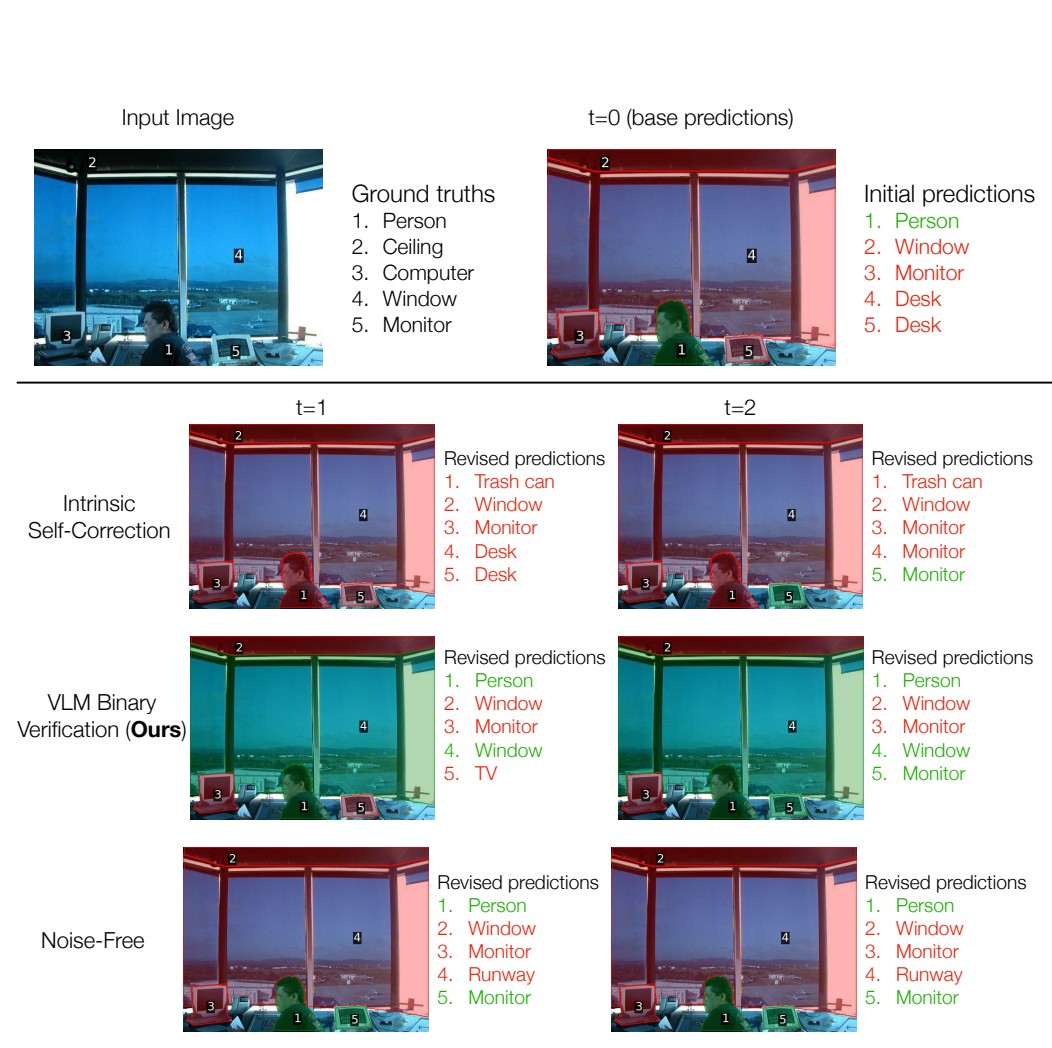

Figure 13: **LLaVA-1.5 qualitative results in ADE20k.** We visualize the predictions of LLaVA-1.5 at time steps from 0 to 2. Intrinsic self-correction fails to identify which predictions are correct/incorrect, while VLM binary verification and Noise-free feedback provide explicit signal on each region, leading to a better chance of correction. From t = 0 to t = 1, we find that VLM might produce different results (object 4) even when receiving the same feedback (VLM binary verification and Noise-free). As explained in Appendix G, in the VLMs forward pass, we draw multiple sequences and take the majority vote as the final responses. For the sake of visualization, we put a bright ID on each object and highlight the incorrect predictions in red and the correct predictions in green.

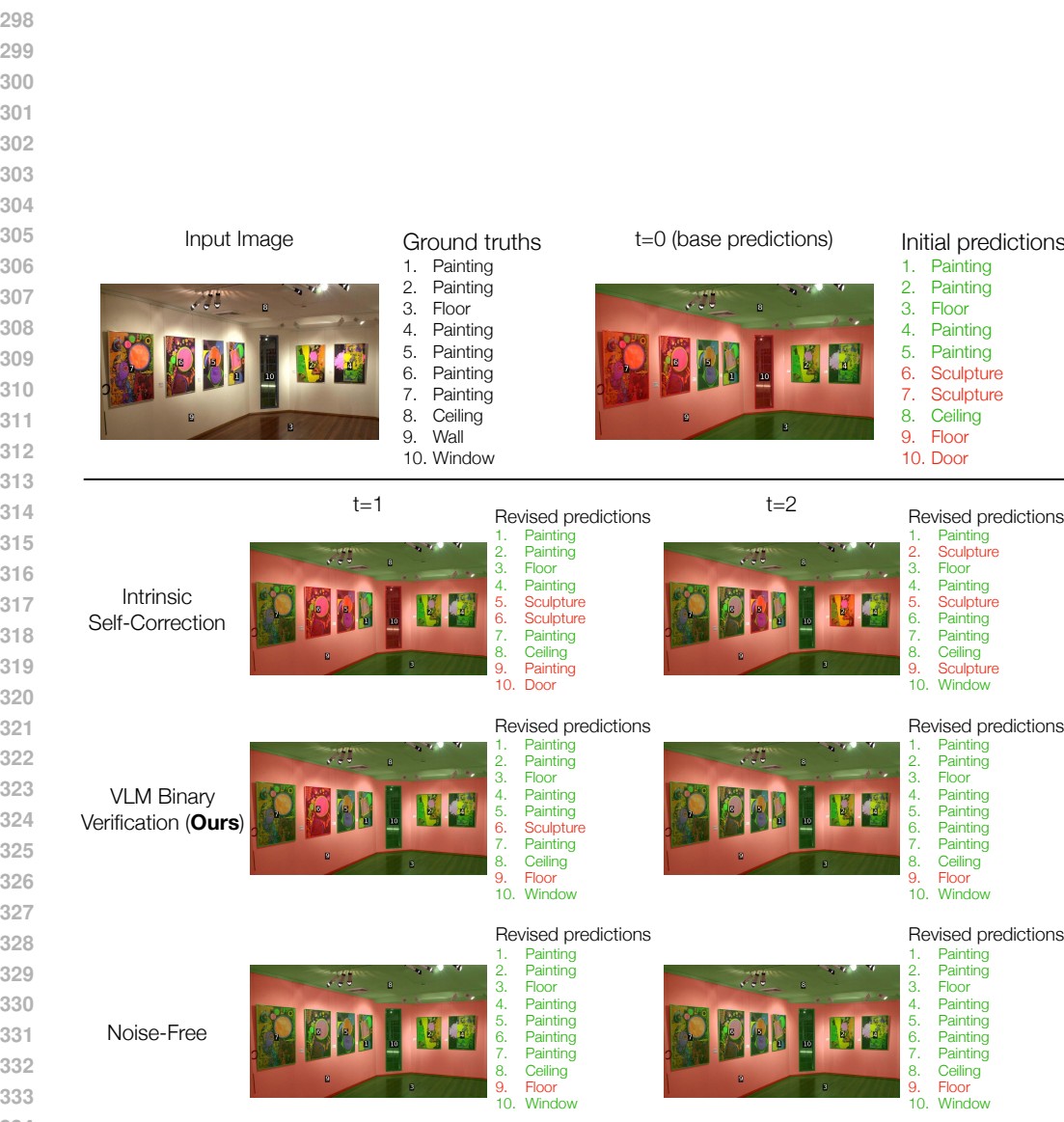

Figure 14: **ViP-LLaVA qualitative results in ADE20k.** We visualize the predictions of ViP-LLaVA at time steps from 0 to 2. Intrinsic self-correction fails to identify which predictions are correct/incorrect, while VLM binary verification and Noise-free feedback provide explicit signal on each region, leading to a better chance of correction. Note that we draw multiple samples in the VLM forward pass, therefore, leading to slightly different results even when the image and query are the same (See Appendix G). For the sake of visualization, we put a bright ID on each object and highlight the incorrect predictions in red and the correct predictions in green.

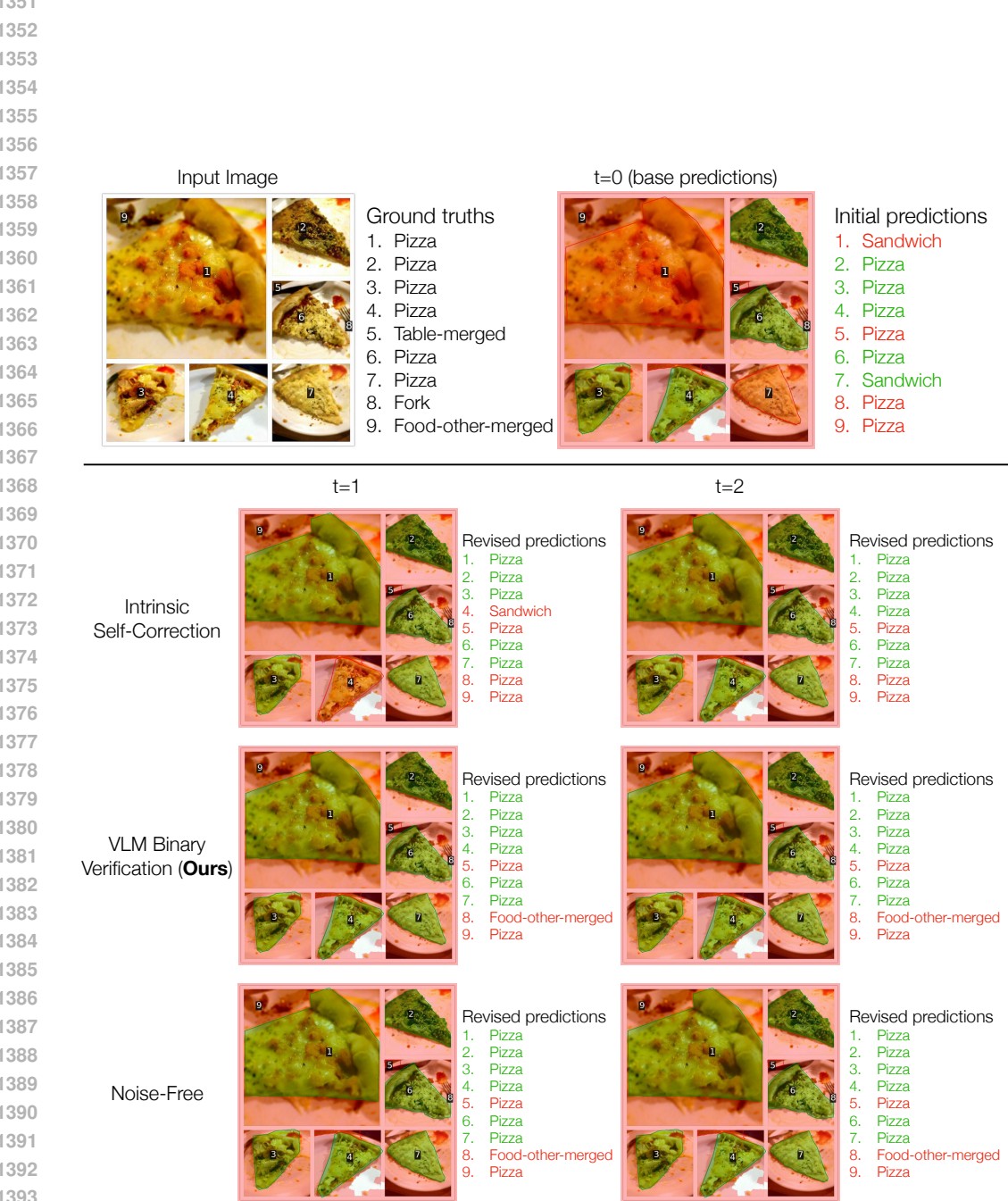

Figure 15: **CogVLM qualitative results in COCO.** We visualize the predictions of CogVLM at time steps from 0 to 2. For the sake of visualization, we put a bright ID on each object and highlight the incorrect predictions in red and the correct predictions in green.

Figure 16: **[Failure case study] LLaVA-1.5 qualitative results in COCO.** All three approaches cannot fix the errors in the initial predictions. For VLM binary verification, from t = 1 to t = 2, the predictions changes from correct (table-merged) to incorrect (cabinet-merged) since the VLM verifier is not perfect and, therefore, providing misleading feedback. Even with the noise-free feedback, LLaVA-1.5 struggle to adjust the predictions. For the sake of visualization, we put a bright ID on each object and highlight the incorrect predictions in red and the correct predictions in green.