# OpenReview forum: "Can Large Vision-Language Models Correct Grounding Errors By Themselves?"
_ICLR.cc/2025/Conference — ICLR 2025 Conference Withdrawn Submission_

### Official Review · Reviewer_Kfzk · 2024-11-03

**Soundness:** 2
**Presentation:** 3
**Contribution:** 2
**Rating:** 3
**Confidence:** 4

**Summary:**

This paper presents a self-correction framework for grounding improvement. It explores the feasibility of using Oracle binary feedback to improve VLMs' semantic grounding performance. Moreover, VLMs themselves are able to provide effective feedback and achieve improvement with prompting. Experiments are conducted for several open-sourced and closed-sourced models across two benchmarks.

**Strengths:**

- The idea of adapting self-correction from LLMs to VLMs for grounding improvement is intuitive.
- Multiple backbones, including LLaVA-1.5, ViP-LLaVA, and CogVLM, are experimented with.
- The experiments are divided into two steps. In the first step, it investigates the Oracle feedback. In the following step, it extends to the VLM-generated feedback. The logic is clear and easy to follow.
- The class label and binary feedback can dramatically improve the grounding performance for 3 architectures, which is encouraging.

**Weaknesses:**

- Even though in the first step, experiments reveal that the label feedback yields much higher improvement than the binary feedback. In the follow-up VLM-generated feedback setting, they do not consider label feedback. Label feedback seems feasible since the masked-out or cropped region can be input into VLM, and it will be a captioning task to obtain a VLM-generated class label. It made the two experiments less coherent, and the final framework should not reach the real upper bound (label feedback).
- As shown in Table 2, each visual mark and RoI crop enhances the model's performance. However, it is concerning that when combining them, the result is worse than either.
- In the main experiment (Table 3 and 4), the baseline intrinsic self-correction is only conducted 3 times, while it is conducted 3-5 times for other methods. It is confusing and possibly unfair.
- The improvement from VLM verification is modest in some cases, such as ViP-LLaVA (0.39%), LLaVA-1.5 (1.91%) even at the 5th timestep. The additional computation does not seem compatible with the desired enhancement.

**Questions:**

- Is it possible to apply VLM-generated label feedback? If yes, what will the method and the corresponding performance be for VLM-generated label feedback? If not, what is the purpose of conducting label and binary feedback experiments in the oracle setting?
- Why can not visual marks and RoI crop work together?
- What will the intrinsic self-correction performance be if you run it 5 times?
- Why does not the VLM verification work well for LLaVA-1.5 on COCO and ViP-LLaVA on ADE20k?

---

### Official Review · Reviewer_yPCF · 2024-11-04

**Soundness:** 2
**Presentation:** 2
**Contribution:** 2
**Rating:** 5
**Confidence:** 3

**Summary:**

This paper explores self-correction in large vision-language models (VLMs) without fine-tuning or modifying the original model parameters/architectures. They choose semantic grounding as the main task and try to answer two questions: first, if the VLMs can understand correct feedback and improve based on it; and second, if the VLMs can provide high-quality binary feedback for themselves. To answer the first question, the authors define two types of feedback (binary feedback and class label feedback) and prompt the VLMs through both textual and visual prompts. They find combining CoT and visual marks is a more effective approach. For the second question, they first evaluate the quality of binary feedback generated by VLMs using F1 scores, and then they iteratively make VLMs perform self-correction up to five rounds. With the experimental results reported using 100 images from ADE20k and COCO, they find intrinsic self-correction brings negative effects in most of the VLMs, but their proposed VLM verification consistently improves model performances for both closed-source and open-source VLMs.

**Strengths:**

1. Self-correction for VLMs is an interesting topic since it does not additional data or sources and may lead to better results.
2. The authors conduct experiments on both open-source and closed-source VLMs, which validate the proposed method more comprehensively.
3. The iterative VLM self-correction consistently improves models' grounding performances in Table 3 and Table 4.

**Weaknesses:**

1. In Table 3 and Table 4, the number of dialogue rounds is set up to 5 and only three open-source VLMs have the results reported with 4 and 5 rounds. The results for GPT-4V and GPT-4o when t=4 and t=5 are missing. Also, when t=5, the three open-source VLMs still get improved compared to the results when t=4. Have the authors tried to keep increasing the number of iterations and see if the performances keep going up or at a specific iteration will decrease?
2. In line 442, "For LLaVA-1.5, the base predictions ... However, adding SoM and using RoI crop results in 11.06 and 19.67, respectively". Have these results reported in any table in the main paper? Also, have the authors tried SoM and RoI crop on ViP-LLaVA and CogVLM?
3. Table 2 shows the quality of binary feedback, and from the table, actually Visual marks+RoI crop is always not the best option for three VLMs. However, the description in line 428 "we use the textual prompts and the visual prompts (e.g., red circles for...) to encourage feedback receiving and use RoI crop when ..." makes me confused. Does it indicate that "red circles"(visual marks) and RoI crop are used in experiments? If so, does it indicate the quality evaluations reported in Table 2 do not necessarily reflect the real effectiveness of visual prompting techniques?

**Questions:**

In lines 346-347, "Table 1 shows...up to 61.06 and 5.18, respectively". Could the authors indicate where to get the "5.18" in Table 1?

---

### Official Review · Reviewer_6ome · 2024-11-04

**Soundness:** 3
**Presentation:** 3
**Contribution:** 3
**Rating:** 6
**Confidence:** 4

**Summary:**

This paper dives into how VLMs can self-correct their mistakes in the area of semantic grounding. Instead of the usual methods like adding more in-domain data or tweaking model architectures, the authors tryto have the VLMs refine their predictions through a process of self-feedback. They focus on whether these models can take feedback effectively and generate useful feedback for themselves.Results in ADE20k and COCO show that this approach can bump up accuracy by quite a bit.

**Strengths:**

* the authors articulate the challenges of semantic grounding well, contextualizing their work within broader research on LLMs and VLMs.
* demonstrated improvements over different datasets, which indicates the practical value of the approach even with noisy feedback
* detailed Analysis: it explores multiple aspects of feedback (oracle vs. self-generated) and the effectiveness of various prompting strategies, enhancing the paper’s impact.
* Requires no additional training data

**Weaknesses:**

* introduce overhead in computational efficiency, which makes it hard to deploy in real-world settings?
* error rate is still high if considering real-world use cases
* is adversarial attack a concern?

**Questions:**

same as above

---

### Official Review · Reviewer_ZAda · 2024-11-04

**Soundness:** 2
**Presentation:** 2
**Contribution:** 3
**Rating:** 3
**Confidence:** 4

**Summary:**

The paper explores an interesting question of whether self correction helps in case of VLMs for semantic grounding tasks. The papers explores good variety of methods to do verification feedback, different prompt techniques to send feedback etc. The paper proposes using the same VLM as the verification step and then use the feedback to correct the previous answer.

The paper tries to answer the following questions :
1. Can VLMs understand the feedback and correct their answers for semantic grounding
2. Are VLMs good at being verifiers , I.e correcting the answers

**Strengths:**

1. The idea is innovative although a similar method exists for LLMs as the paper explores different techniques Possible ways to take feedback  and do verification.

**Weaknesses:**

1. Although the idea of verification is great, the paper mentions using a method that can isolate the object of interest in the verification step by cropping and other techniques which potentially reduces the complexity for the verification and is not a correct way for doing self correction implementation. Instead they should use the same image for verification step as well.
2. Numbers in the tables and section 4 doesn’t add up, some statements like the caption in table 2 sound too vague (“choices of visual prompting techniques should be tailored to the specific VLMs”).

**Questions:**

1. Can you use the same image without any modifications for the verification step and show the results ?
2. Your table 1 and 2 needs change for format to better explain the contents for the reader, also correct the numbers if they are wrong.
3. Would really suggest that you to do a good rewrite of the paper to better explain the things as the idea and approach seems very good.

---

### Note · Authors · 2024-11-17

I have read and agree with the venue's withdrawal policy on behalf of myself and my co-authors.